# The Effects of Yogurt Supplementation and Nutritional Education on Malnourished Infants: A Pilot RCT in Dhaka’s Slums

**DOI:** 10.3390/nu15132986

**Published:** 2023-06-30

**Authors:** Kaniz Jannat, Kingsley Emwinyore Agho, Sarker Masud Parvez, Mahbubur Rahman, Russell Thomson, Mohammed Badrul Amin, Dafna Merom

**Affiliations:** 1Centre for Research in Mathematics and Data Science, School of Health Sciences, Western Sydney University, Locked Bag 1797, Penrith, NSW 2751, Australia; kaniz.jannat@gmail.com (K.E.A.); russell.thomson@westernsydney.edu.au (R.T.); d.merom@westernsydney.edu.au (D.M.); 2Environmental Interventions Unit, Laboratory of Food Safety and One Health, Infectious Disease Division, Laboratory Sciences and Services Division, International Centre for Diarrhoeal Disease Research, Bangladesh (icddr,b), Dhaka 1212, Bangladesh; parvez@icddrb.org (S.M.P.); mahbubr@icddrb.org (M.R.); badrul.amin@icddrb.org (M.B.A.); 3Children’s Health and Environment Program, Child Health Research Centre, The University of Queensland, South Brisbane, QLD 4101, Australia

**Keywords:** yogurt, child growth, LMICs, Bangladesh, RCT

## Abstract

Our objective was to quantify the effects of yogurt supplementation and nutrition education over three months on the linear growth of infants at risk of stunting. We conducted a three-arm pilot randomized controlled trial: (1) nutrition education for mothers; (2) nutrition education plus a daily yogurt supplement (50 g) for the index child; and (3) usual care (control). Dyads of children aged 4–6 months and at risk of stunting [length-for-age z-score (LAZ) ≤ −1 SD and >−2 SD] and their mothers with ≤10 years of education were eligible for the study. Participants were recruited from five slum areas in Dhaka, Bangladesh. Intention-to-treat (N = 162) and complete-case analyses (N = 127) showed no between-group statistically significant differences in LAZ or weight-for-age (WAZ). However, the yogurt group showed greater change in linear growth compared to the control (LAZ: mean difference 0.20, 95% CI: −0.06, 0.47, *p*-value 0.13), which was also slightly greater than the education-only group. Children in the yogurt plus group were five times (95% CI: 0.80, 31.80, *p*-value 0.09) more likely to meet the minimum dietary diversity (MDD) score compared to the control. A 3-month follow-up of this pilot study did not demonstrate that yogurt was beneficial to linear growth. However, there were encouraging trends that merit replication of the intervention with larger samples and longer follow-ups.

## 1. Background

Child undernutrition is a major public health concern, contributing substantially to child mortality and disease burden in low- and middle-income countries (LMICs) [1]. Stunting, a form of chronic undernutrition, is associated with increased child morbidity and mortality [2] and is associated with long-term consequences, including a greater risk of developing non-communicable diseases [3]. Poor school performance, reduced economic attainment, lower productivity, and unfavourable maternal reproductive outcomes are also related to stunting [4]. In 2020, stunting affected 22.0% of children under five globally, and in Bangladesh, the situation was even more severe; 30.2% of children were stunted [5]. A stark difference in stunting between slum and rural regions was also demonstrated (50% vs. 36%, respectively) [6].

While child stunting is a multifactorial condition, repeated infections and nutritional insufficiencies were identified as its immediate causes [7]. There are several underlying and basic risk factors that were consistently found to be associated with undernutrition and stunting globally, such as low levels of household income, food insecurity, unhealthy household environments (e.g., poor sanitation and air quality), inadequate health services, and low maternal education [2,8,9]. 

Different approaches have been trialled to address this complex problem of child undernutrition or stunting. Generally, nutrition-specific interventions have been implemented for years to address stunting. Different systematic reviews indicated that basic nutritional education was effective at lowering undernutrition among children to a certain extent; when complementary or supplementary foods were added to nutritional education, the results improved [10,11,12]. The types of complementary foods used in different trials varied greatly, from common household nutritious food items (e.g., eggs, milk) to ready-to-use foods like lipid-based nutrient supplements and micronutrient supplements. Nevertheless, food quantity or quality was not able to fully explain the linear growth failure in young children [13]. 

Impaired functionality of the small intestine, known as environmental enteric dysfunction (EED), is now believed to be the underlying aetiology of nutritional deficits in children living in unsanitary conditions like slums [3,14,15]. Substantial derangement in gut microbial composition, a potential predictor of EED [16], has been found in malnourished children living in low-income settings [17,18,19]. Improving gut health is an emerging approach that could minimize the adverse consequences of EED and improve child nutritional status in LMICs [20]. In the review by Tickell, Atlas, and Walson, five out of ten trials that evaluated dietary supplements (protein, micronutrients, probiotics, and naturally occurring novel supplements) for EED showed an increase in child growth. Only a handful of trials have so far tried gut microbiota-targeted interventions (probiotic, prebiotic, and symbiotic interventions) to improve growth outcomes in children living in LMICs [21]. In Heuven’s systematic review, five out of eleven studies demonstrated beneficial effects of probiotics on one or more growth parameters.

A fermented food can be considered a probiotic food when it contains characterized cultures and viable microbes at the time of consumption [22]. Fermentation increases the nutritional value of food by adding beneficial bacteria, bioactive compounds, vitamins, and enzymes [22]. The health benefits of fermented foods have been validated in recent years, focusing on nutrition related conditions such as obesity, diabetes, cardiovascular disease, gut health, and cancer [23]. Currently, there is a paucity of trials that explore the impact of fermented foods like yogurt on child growth among young children [24,25]. 

Commercially produced probiotics are supplements, whereas fermented foods are part of a meal. Yogurt is one of the most bioactive foods consumed by humans. The nutritional value of yogurt is high relative to its cost. In Bangladesh, yogurt is a traditional fermented dessert item produced both commercially and at the household level [26]. Its promotion as a complementary food is therefore potentially feasible in low-income households where food insecurity is present. 

To our knowledge, no study to date in Bangladesh has tested the efficacy of yogurt on infant growth as early as six months of age, when children are at increased risk of exposure to pathogens affecting their gut health [27]. In this pilot study, we hypothesize that yogurt supplementation along with nutrition education for continuing breastfeeding and adequate complementary feeding could enhance growth among infants at risk of stunting in Dhaka slums, where there is a high risk of EED due to unsanitary conditions [28], a lack of dietary diversity influenced by food insecurity [29], and poor feeding practices compounded by the high prevalence of low-educated mothers [30].

## 2. Methods

### 2.1. Study Design

A pragmatic pilot randomised controlled trial was conducted with three arms: (a) nutrition education; (b) nutrition education and yogurt supplement; and (c) the usual-care (control) group. The study was conducted in five slum areas in Dhaka city: *Mohakhali*, *Nakhalpara*, *Korail*, *Badda*, and *Kalshi* (Figure 1). Considering the nature of the intervention, which included daily home visits to deliver yogurt, we chose areas closest to the research centre. The study was planned for six months with two follow-up assessments: one at three months and another at six months. However, due to the COVID-19 pandemic, we ended up with only one follow-up assessment after three months. Bangladesh was under lockdown for two months when the study children should have received yogurt intervention daily. Since we were unable to continue with our field activities to distribute yogurt, we abandoned the six-month follow-up.

### 2.2. Ethical Approval and Trial Registration

The study protocol was approved by the Ethical Review Committee of the International Centre for Diarrhoeal Disease Research, Bangladesh (icddr,b) (Approval Code: PR-19062; Approval Date: 19 June 2019) and the Human Research Ethics Committee at Western Sydney University (Approval Code: H12719; Approval Date: 23 July 2018). The trial was registered at *ClinicalTrials.gov* with trial registration number NCT04067284. 

### 2.3. Participants and Recruitment

Children aged 4–6 months and at risk of stunting (length-for-age z-score, LAZ ≤ −1 SD and >−2 SD), and their mothers’ reported education of less than ten years were eligible for the study. Children with moderate to severe malnutrition, congenital abnormalities, or any chronic conditions at screening were excluded from the study. 

Field research assistants (FRAs) identified potential eligible children from the vaccination database at the Expanded Program of Immunization (EPI) Centre, Mohakhali, Dhaka, Bangladesh. The FRAs contacted the parents by phone, explained the study in brief, and obtained verbal consent to visit their households for anthropometric measurement of the index children to complete the screening process. If agreed upon, the FRAs then travelled to the households and measured the recumbent length of the index children and plotted the value on the growth chart. Parents of the children who met the eligibility criteria were invited to consent to their children’s participation in the study. FRAs handed the participant information sheet to the parents; it was explained or read to them if needed; FRAs answered all queries and concerns and obtained informed written consent from the fathers as household heads.

### 2.4. Randomisation and Masking

The block randomisation method was applied using the STATA command to randomise participants into any of the three study arms in equal proportion. A randomisation sheet with a block size of four was generated by a co-investigator (KA) at Western Sydney University, Australia. FRAs enrolled the study participants daily, and at the end of the day, the student investigator (KJ) did the final allocation of the participants using the pre-allocated randomisation sheet. Allocation was concealed from research staff until the baseline assessment was complete. Since the intervention included visible components, FRAs were not blinded for subsequent measurements, nor were the participants.

### 2.5. Intervention Groups

#### 2.5.1. Intervention-1: Nutrition Education Enhanced with Self-Monitoring Using a Pictorial Calendar (Education-Only Group)

We have taken the behavioural recommendations from previous successful nutrition interventions in Bangladesh, namely the WASH Benefits Bangladesh trial [32]. Health promoters (a different group of FRAs) launched the nutrition education intervention in a pre-arranged face-to-face monthly household meeting. Each session lasted for about an hour, allowing mothers to attend to household commitments. As in the WASH Benefits study [32], we used illustrated flip charts to convey messages to mothers and members of their households. Educational contents included continued breastfeeding, food groups, a balanced diet, dietary diversity in complementary feeding, the amount, consistency, and frequency of feeding, feeding during sickness, safe water, and handwashing with soap before feeding. Different sessions included different activities to make them unique, more appealing, and more engaging for mothers. In addition to flip charts, we included quizzes, flashcard games, and storytelling components in different sessions. 

Pictorial calendar: To enhance compliance with optimal complementary feeding, we designed a pictorial calendar to assist mothers in self-regulating and monitoring their feeding practices. This addition to the study follows the findings that communication of knowledge may not be enough to elicit behaviour change [33]. The Social-Cognitive approach to behaviour change highlights the importance of self-monitoring and self-regulation as a step to building self-efficacy [34,35], a concept that is described as a critical determinant of health-related behaviour change [36]. There are several examples of increased effectiveness of interventions where self-regulatory behaviour-change techniques, such as setting goals and self-monitoring, were used [37]. However, these studies were conducted in developed countries where participants were able to read and write and could make informed choices from a variety of options. We therefore designed a visual aid for self-monitoring in the form of a pictorial calendar to assist low-educated mothers to self-regulate their child feeding behaviours and increase adherence to recommended child feeding practices.

#### 2.5.2. Intervention-2: Yogurt Supplement plus Nutrition Education Enhanced with Self-Monitoring Using pictorial Calendar (Yogurt-Plus Group)

In addition to the above nutritional education activities, this group received a cup of yogurt (50 g) every day delivered to the households by an FRA. Yogurt was introduced to the children after one week of regular complementary feeding following the first nutrition education session. Children were, on average, 190 days old when yogurt was first introduced. The FRAs tried to ensure that the yogurt was fed to the index child in his/her presence; if the yogurt was left for later feeding, mothers were asked to feed it within two hours of delivery. 

#### 2.5.3. Usual Care (Control Group)

This group received no intervention from the research team and represented the current recommended practices among healthcare providers. Depending on the needs of mothers and children under five years of age, local governments, non-profit organizations (NGOs), or private facilities were available to provide primary healthcare and nutrition rehabilitation or counselling. 

### 2.6. Intervention Materials

#### 2.6.1. Pictorial Calendar

The pictorial calendar contained coloured stickers that represented different food groups: energy-yielding foods were marked with yellow stickers, body-building foods with red stickers, protective foods with green stickers, and blue stickers to indicate breastfeeding. Mothers had to use coloured stickers every day to reflect what food groups were fed to the index child. Mothers were encouraged to feed at least one food item from each of the three basic food groups: carbohydrate and fat, protein, and fruits and vegetables.

#### 2.6.2. Making Yogurt, Distribution, and Quality Control

The FRAs made yogurt from full-cream milk powder commonly available in the local market and yogurt culture without adding any sugar or flavour. Yogurt starter kit containing classic yogurt culture strains, *Lactobacillus bulgaricus* and *Streptococcus thermophilus,* was imported from Australia. It was manufactured by Green Living Australia. At the research centre, yogurt was prepared every day in hygienic conditions and stored overnight at 3–5 degrees Celsius. Yogurt was transported to households in disposable cups with spoons using cool boxes, maintaining a temperature between 4 and 10 degrees centigrade. The prepared yogurt was tested at the Laboratory of Food Safety and One Health, icddr,b twice a week for identification and enumeration of the culture bacteria according to the procedures described earlier [38]. The average concentration of *Streptococcus thermophillus* was 2.6 × 10^8^ CFU/g and *Lactobacillus delbrueckii* subsp. *Bulgaricus* was 1.1 × 10^8^ CFU/g, which met the recommended level for daily consumption [39]. The supplied yogurt (50 g) provided ~32 calories of energy, which was ~5% of the daily requirement of six month to nine-month-old children. 

### 2.7. Outcome Measures and Covariates

Primary outcome: two anthropometric measures were used to assess intervention effectiveness: change in mean LAZ and weight-for-age z-score (WAZ) from baseline to 3-month follow-up. FRAs were trained and standardized according to FANTA and World Health Organization (WHO) guidelines for anthropometric measurements. They measured the recumbent’s length and weight with minimal clothing using standardized techniques [40,41]. Three repeated measurements were recorded within an error limit of ≤0.5 cm for length and 0.1 kg for weight. Recumbent length was measured with a Seca 417 infant length board with a precision of 1 mm, child weight was assessed with the Seca 874 scale with 0.1 kg precision. Measurement scales were calibrated at the households each time before taking measurements. 

Secondary outcome: We used three measurements that indicated dietary diversity based on a 24 h recall. We used the WHO infant food frequency questionnaire (FFQ), piloted and adapted by the WASH Benefits study [32]. We calculated the indicators based on “Indicators for assessing infant and young child feeding practices: Definitions and measurement methods” [42]. 

(1)Minimum dietary diversity (MDD): the proportion of children who received food from five or more food categories out of the eight food groups on the previous day. To keep the scores comparable between the groups, the supplied yogurt was not considered while estimating the MDD score for the yogurt-plus group.(2)Minimum meal frequency (MMF): the proportion of children who received solid, semi-solid, or soft foods the minimum number of times or more on the previous day. The minimum acceptable number is two or more times until eight months and three or more times thereafter.(3)Minimum acceptable diet (MAD): the proportion of children who meet both MDD and MMF scores.

Covariates: Most covariates, such as child gender, birth outcome, birth order, parents’ education, and household size, were measured at baseline before randomisation. Two modules regarding household food insecurity and wealth assessment, adapted from the WASH Benefits study [32], were assessed during a three-month follow-up, to minimize the length of the assessment rounds. 

Adverse outcomes: Severe adverse events were defined as the hospitalization of index children for any reason. If any index child was found hospitalized during routine study activities or reported to the study staff, a severe adverse event report was submitted to the icddr,b Ethics Review Board within 24 h of the first contact with the research team. After following up on the case, a report was submitted stating the final outcome. 

### 2.8. Monitoring Intervention Adherence

Yogurt consumption was monitored every day, with FRAs reporting the amount of consumption on a log sheet. We defined adherence as the consumption of at least half a cup (≥25 g) of yogurt on any given day. The number of days that yogurt was delivered to each child was used as the denominator for calculating adherence.

Calendar pages were retrieved at the end of the month and checked for completeness. The next household meeting was adapted according to the individual mother’s needs, such as a briefing regarding right use by addressing the difficulties in filling out the calendar (e.g., they did not understand how to select coloured stickers and attach them to the calendar according to date). 

### 2.9. Data Collection and Procedures 

The FRAs worked in three groups: a survey group, a health promotion group, and a yogurt distribution group. Screening and recruitment were done by all FRAs to make enrolment faster. Recruitment was completed in four months, from the end of September 2019 to the end of January 2020. The survey group of FRAs conducted baseline and three-month follow-up assessments. Baseline assessment was done within one week of enrolment; three-month follow-up assessment was done 90 days (±1 week) after baseline assessment. Health promoters conducted the first nutrition education meeting within one week of baseline assessment, which was around ±1 week of 6-month of child age. Yogurt distribution started one week after the first nutrition education meeting and continued daily. 

### 2.10. Sample Size and Statistical Analysis

The sample size was calculated to detect minimal mean differences between any intervention and control in a standardized LAZ of 0.3 with a power of 90% and a type 1 error of 5%. Since we aimed to recruit children at risk (LAZ of −1 SD from the mean of 0), we hypothesized that after six months of intervention, at the age of twelve months, the mean z-score of the usual care group will remain unchanged, whereas the education-only group will improve their z-score from −1 to −0.49 while the mean z-score of yogurt-plus group will further improve to −0.29, assuming a similar standard deviation of 0.67 in all groups, thus, the required sample would be 120 (40 in each arm). Considering a 15% attrition rate over two years in a large RCT in rural Bangladesh [43] and a rapid turnover in the urban slum population, we aimed to recruit 162 children (54 per arm) to allow for a 25% loss to follow up after six months.

Demographic and other baseline covariates were compared between groups using *t*-tests and chi-square tests, as appropriate. A principal components statistical procedure on household assets, water sanitation facilities, and household building materials was used to estimate the household wealth index factor score. The household wealth index factor score was divided into wealth quintiles [44]. 

Both intention-to-treat and complete-case analyses (e.g., the complete-case analysis included participants without missing data on the variables of interest) were conducted to assess the efficacy of the intervention [45,46]. To measure the association between intervention and study outcomes, we applied linear and logistic mixed models for panel data [47]. A mixed-effects model was used to adjust the variability that was unexplained by the predictors; the participants were treated as a random effect, whereas treatment and covariates were treated as fixed effects. Linear regression was used for continuous outcomes (anthropometric measurements), and logistic regression was used for analysing categorical outcomes (e.g., MDD, MMF, MAD). For the ITT analysis, we imputed the missing data using the multiple imputation method. We checked the association of child gender, mode of birth, parent’s education, food security, and household wealth with the main outcomes by univariate regression analysis since these are known predictors of our main and secondary outcomes. As there were no significant differences between the adjusted and unadjusted models, only the unadjusted results are reported in this paper. Analyses were carried out with STATA (version 13.0).

## 3. Results

FRAs screened 710 children; 527 children did not meet the eligibility criteria, and 162 children were enrolled in the study. 128 children out of 162 (79%) completed the three-month follow-up. We could not reach 34 children (21%) for a three-month follow-up, primarily due to the discontinuation of the study for the COVID-19 pandemic (9%), and seven participants (4%) withdrew from the intervention since they were not willing to receive the intervention any further (Figure 1). Children who did not complete the three-month follow-up had better mean weight and WAZ.

At baseline, there was no significant difference in any of the study covariates except for the history of a Caesarean section (Table 1). Caesarean section birth was more common in the education-only group than in the control and yogurt-plus groups (control 33.3%, education-only 55.6%, yogurt-plus 37.0%; *p*-value 0.04).

The average study duration for all participants was 91.2 days (SD 3.7, range 90–114 days); control was 91.0 days, education-only was 92.5 days and yogurt-plus was 91.2 days. On average, yogurt was consumed 86.1% of the days it was distributed (95% CI: 80.6, 91.6). 

According to the ITT analysis (N = 162), there were no significant changes in mean LAZ or WAZ between the yogurt-plus group and the control (Table 2). Both intervention groups showed better growth parameters compared to control, but the between group mean differences were not statistically significant; yet the improvement in growth measures in the yogurt-plus group tended to be larger than the education-only group (mean difference in LAZ: education-only: 0.09, 95% CI: −0.17, 0.34 *p*-value 0.51, yogurt-plus: 0.13, 95% CI: −0.12, 0.38, *p*-value 0.31; mean difference in WAZ: education-only: 0.08, 95% CI: −0.14, 0.30 *p*-value 0.46, yogurt-plus: 0.13, 95% CI: −0.05, 0.32, *p*-value 0.15). The complete-case analysis showed (Table 2 and Figure 2) that at 3-month follow up, children in the education-only group had an increase in LAZ by a mean difference of 0.13 (95% CI: −0.13, 0.39, *p*-value 0.32) and yogurt-plus group by 0.20 (95% CI: −0.06, 0.47, *p*-value 0.13) compared to control children. There was an almost similar increase in WAZ in both intervention groups compared to control (mean difference: education-only: 0.06, 95% CI: −0.12, 0.25 *p*-value 0.50; yogurt-plus: 0.05, 95% CI: −0.14, 0.24, *p*-value 0.59).

On the ITT analysis, the infant food frequency survey revealed that children in the education-only group were 2.44 times (95% CI: 0.45, 13.21, *p*-value 0.30), and children in the yogurt-plus group were 5.0 times (95% CI: 0.79, 31.80, *p*-value 0.09) more likely to meet the MDD score compared to the control children. The complete-case analysis also showed a similar result (Table 3).

There were six events of hospitalization due to diarrhoea, vomiting, and/or fever; three children were from the education-only group and three from the yogurt-plus intervention group. None of the cases were diagnosed as resulting from yogurt consumption.

## 4. Discussion

To the best of our knowledge, this is the first randomized controlled trial that tested the effects of yogurt supplementation along with nutritional education on the growth parameters of 6-month-old infants in Bangladesh. We found no significant between-group differences in mean LAZ or WAZ in either ITT or complete-case analyses. However, in both ITT and complete-case analyses, we have seen a consistent tendency to demonstrate better growth outcomes (mean length and LAZ) in favour of the yogurt-plus group compared to the control group, which was slightly greater than the education-only group. Further, in both ITT and complete-case analyses, a 24-h recall of food consumption revealed that children in both the yogurt-plus and education-only groups were more likely to meet the MDD score compared to the control, with the yogurt-plus group presenting a greater effect.

Three possible explanations for the non-significant effects of the intervention can be considered: first, the study was too short to demonstrate growth benefits from yogurt consumption. To estimate growth trajectories, at least two measurements at least two months apart are required; however, longer intervals and additional measurements are desirable [48]. Second, lack of statistical power for the smaller effect size detected in this study on LAZ (0.20) as opposed to the anticipated effect size of 0.35 following a 6-month intervention that guided the sample size calculation. Third, the dose of yogurt was not enough to exert its potential effect on growth parameters among infants ‘at risk’ of stunting living in slums, since the pathways the intervention was expected to influence growth (through improved gut health) are also influenced by a complex interplay of different factors at all levels [49], such as poor maternal health, birth outcome, and unsanitary conditions.

Limited studies employed gut microbiota-targeted interventions to improve child growth in LMICs. The systematic reviews conducted by Heuven et al. and Donovan et al. included seven probiotic and yogurt intervention studies conducted in LMICs. These studies generally observed an impact on weight gain; however, their population was different (hospitalized, stunted, or healthy) than the present study. Two studies, one in India [50] and one in China [51] used yogurt interventions delivered to older malnourished children of 2–5 years; both studies showed an increase in height and weight in favour of the intervention. However, in both studies, intervention was deployed for a longer duration (6–9 months) compared to the present study. At-risk children living in slums might require longer intervention, which might be one of the important reasons for not achieving significant improvement in growth parameters in this study.

For low-educated mothers to achieve better feeding practices, we choose nutrition education as a minimal intervention along with yogurt supplementation. Panjwani et al. in their systematic review presented the pooled effect of linear growth in children aged 6–23 months in LMICs from six RCTs; nutrition education regarding complementary feeding alone was able to increase LAZ by a mean difference of 0.13 (95% CI: 0.01, 0.24), exactly the effect seen in the education-only group. We saw a larger mean difference in LAZ in the yogurt-plus group, which is in-line with the findings that combined education with nutritional supplementation was better at improving growth [11,52]. Our findings, therefore, support that the nutrition education intervention was effective as in other RCTs, and the lack of statistical significance was probably due to our small sample. Nutrition education alone to improve complementary feeding practices were studied extensively in different countries, including Bangladesh [32,53,54,55]. A meta-analysis of 83 studies of different delivery platforms reported a higher likelihood of achieving the minimum dietary diversity score of 2.34 times (95% CI: 1.17, 4.70, *n* = 4) among mother/peer group platforms [49], an effect that is in line with the effect seen for the education-only group in this trial. The greater effect we saw in the yogurt-plus group on meeting the recommendation for optimal feeding practice may be explained by increased frequency of contact with the study team; a systematic review on the effectiveness of home visiting programs on child outcomes revealed that frequent home visits were associated with better outcomes [56]. Given the above, the better feeding practices demonstrated in the yogurt-plus group (significant at the 10% level) could be due to a cumulative effect of face-to-face monthly nutrition education, self-monitoring of child feeding practices using the pictorial calendar, and a passive effect of daily home visits by the research staff for delivering yogurt.

Our objective was to reduce gut inflammation through yogurt consumption to enhance child growth, a novel strategy to address malnutrition. A systematic review by Harper et al. considered intestinal inflammation as one of the five domains to describe the complex mechanism by which EED may contribute to stunting [13]. As indicators of intestinal inflammation, alpha-1 antitrypsin (AAT), myeloperoxidase (MPO), and neopterin (NEO) are frequently used along with other indicators [57,58]. Cut-off values for most of these biomarkers have not been established in children with malnutrition [59], so any reduction might be considered an indication of a decrease in gut inflammation. In the WASH Benefits study, faecal NEO concentration was significantly reduced in the nutrition intervention arm (lipid-based nutrient supplements and nutrition education) compared to controls in the 3- and 14-month follow-up [32]. In this study, we measured AAT, MPO, and NEO concentration before and after the yogurt intervention (data not included) and found a small, but promising decrease in NEO concentration (21% lower) among children in the yogurt-plus group compared to controls, which was not seen in the education-only group, but again, the between-group mean change in the NEO biomarker was not statistically significant [60].

### Strengths and Limitations

The execution of the study was robust; it was implemented in collaboration with the icddr,b, one of the world’s leading global health research institutes. We used the infant food frequency questionnaire, a standard measurement tool recommended by the WHO that is not free from recall or social desirability bias. However, it should have affected all participants equally, regardless of allocation. We prepared the yogurt in a controlled environment, but not in the laboratory. This implies that if basic food hygiene is maintained during yogurt preparation, it should be safe for children. The intervention looks promising from the perspective of scalability.

As a pilot study, the sample size was not calculated for estimating a small effect size. A sudden study closure due to the COVID-19 outbreak reduced the sample even further. We could not reach the scheduled assessment at six months, which may have precluded establishing the association between yogurt consumption and child growth. Improved dietary diversity in the yogurt-plus group might have influenced child growth in addition to the yogurt supplement. We were unable to isolate the effects of yogurt, which was a limitation of the study; a comparable arm receiving milk or other carbohydrates providing equal calories would have better isolated the influence of yogurt. Other methodological limitations include the inability to blind the participants or the research observers for subsequent visits due to the visible component of the intervention. Given the primary outcome measures were all objective and strict protocols were followed, such performance bias is unlikely.

## 5. Conclusions

In this three-month trial, based on a partial sample, we saw a better trajectory of growth among children in the yogurt intervention group compared to the education-only or control groups. It was not possible to determine whether this improvement was solely due to yogurt consumption. Yogurt, however, can be a cost-effective intervention with immense possibilities to improve growth and development in children through multiple pathways, which warrants further testing. Yogurt was found safe for infants and was well accepted as a complementary food. It is possible to increase yogurt consumption through training and household engagement.

## Figures and Tables

**Scheme 1 nutrients-15-02986-sch001:**
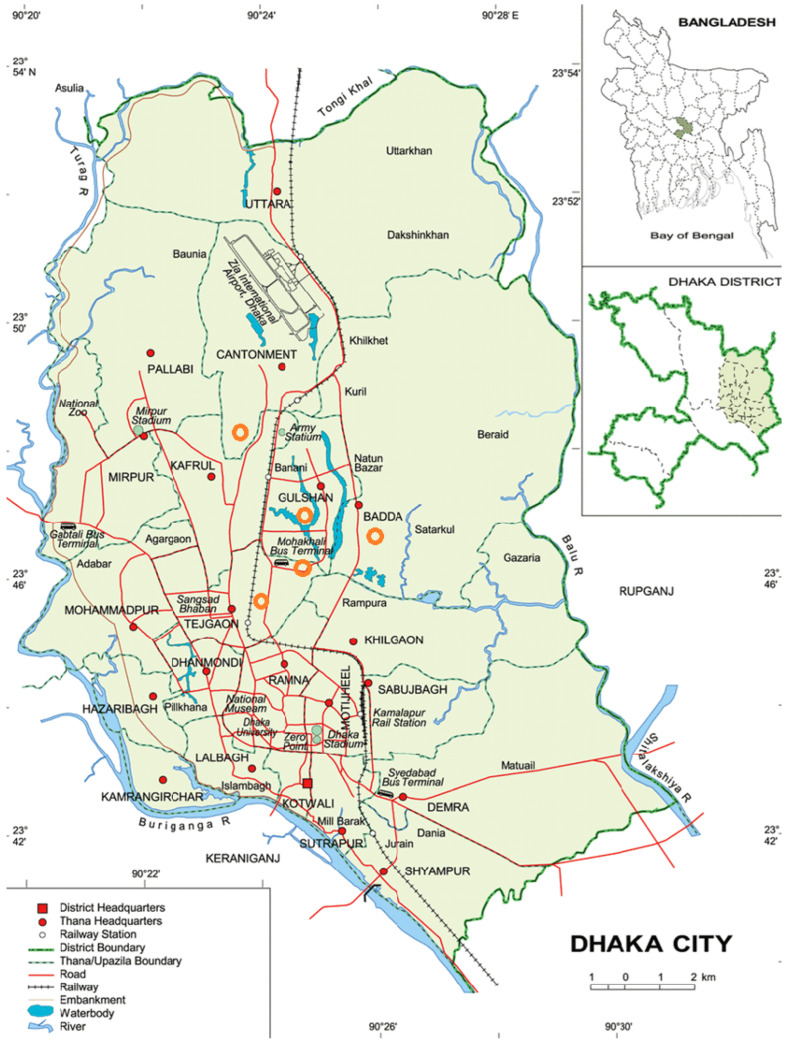
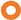
 Study sites in Dhaka, Bangladesh [31]; study sites are marked on the original image available via license: Creative Commons Attribution 4.0 International.

**Figure 1 nutrients-15-02986-f001:**
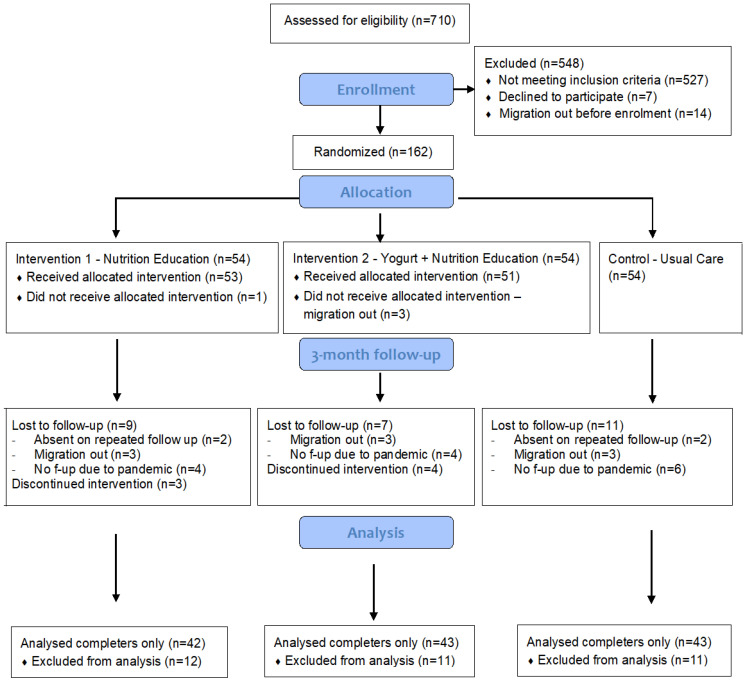
Study flow diagram.

**Figure 2 nutrients-15-02986-f002:**
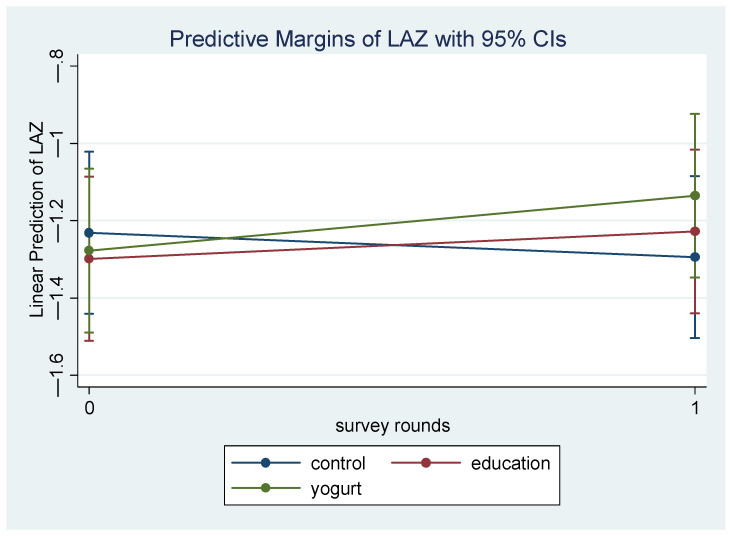
Length-for-age z-score (LAZ) trajectory at two assessment periods in complete-case analysis.

**Table 1 nutrients-15-02986-t001:** Participants characteristics at baseline and follow up by study groups.

Indicators		Control *n* = 54 (%)	Education-Only *n* = 54 (%)	Yogurt-Plus *n* = 54 (%)
Child age in days at baseline survey (mean, 95% CI)		180 (179, 182)	180 (179, 181)	180 (178, 181)
Child gender (male)		53.7	50.0	50.0
Type of delivery: caesarean section		33.3	55.6	37.0
Birth order	First	42.6	44.4	37.0
	Second	33.3	38.9	40.7
	Third	18.5	13.0	13.0
Mother’s age in years (mean, 95% CI)		25 (23, 26)	24 (23, 25)	25 (23, 26)
Father’s education (mean, 95% CI)		5.1 (4.0, 6.1)	5.7 (4.6, 6.8)	6.1 (5.0, 7.2)
Mother’s education (mean, 95% CI)		4.8 (4.0, 5.5)	5.1 (4.4, 5.8)	4.8 (4.1, 5.6)
Household size (mean, 95% CI)		4.7 (4.2, 5.2)	4.8 (4.2, 5.4)	5.1 (4.5, 5.8)
Shared improved sanitation facilities (pour-flush latrine to piped sewer system)	Shared with <10 households Shared with ≥10 households	74.1 25.9	68.5 31.5	75.9 24.1
Sources of drinking water (municipal supply)		98.2	98.2	100
Continued breastfeeding		100	88.9	94.4
		* *n* = 43 (%)	* *n* = 42 (%)	* *n* = 43 (%)
Prevalence of food insecurity	Food secure	51.2	40.5	55.8
	Mildly food insecure	9.3	7.1	7.0
	Moderately food insecure	28.0	31.0	30.2
	Severely food insecure	11.6	21.4	7.0
^†^ Wealth	Quintile 1 (poorest)	20.9	23.8	18.6
	Quintile 2	27.9	21.4	9.3
	Quintile 3	16.3	21.4	23.3
	Quintile 4	18.6	21.4	18.6
	Quintile 5	16.3	11.9	30.2

* Full sample was not reached at the three month follow-up; ^†^ Wealth index was calculated using Principal Component Analysis.

**Table 2 nutrients-15-02986-t002:** Effect of nutrition education-only and yogurt-plus interventions on child growth.

Anthropometry Indicators	ITT Analysis N = 162				Complete-Case Analysis N = 127			
	Baseline Mean	3-Month Follow Up Mean	^‡^ Mean ∆	(95% CI)	Baseline Mean	3-Month Follow Up Mean	^‡^ Mean ∆	(95% CI)
Length (cm)								
Control	63.87	68.15	Ref.	-	63.86	67.95	Ref.	-
Education-only	63.72	68.24	0.24	−0.37, 0.84	63.79	68.25	0.37	−0.23, 1.00
Yogurt-plus	63.81	68.38	0.29	−0.29, 0.88	63.76	68.29	0.45	−0.16, 1.05
Length-for-age z score (LAZ)								
Control	−1.25	−1.23	Ref.		−1.23	−1.29	Ref.	-
Education-only	−1.29	−1.19	0.09	−0.17, 0.34	−1.30	−1.23	0.13	−0.13, 0.39
Yogurt-plus	−1.24	−1.09	0.13	−0.12, 0.38	−1.28	−1.14	0.20	−0.06, 0.47
Weight (kg)								
Control	7.04	7.84	Ref.		6.84	7.67	Ref.	-
Education-only	6.92	7.80	0.07	−0.12, 0.27	6.83	7.74	0.07	−0.09, 0.23
Yogurt-plus	6.86	7.77	0.11	−0.05, 0.27	6.90	7.79	0.05	−0.11. 0.22
Weight-for-age z-score (WAZ)								
Control	−0.75	−0.83	0		−0.96	−0.99	0	-
Education-only	−0.83	−0.83	0.08	−0.14, 0.30	−0.97	−0.94	0.06	−0.12, 0.25
Yogurt-plus	−0.89	−0.84	0.13	−0.05, 0.32	−0.84	−0.83	0.05	−0.14, 0.24

^‡^ Mean difference estimated using Linear Mixed Effect Model; missing values generated using multiple.

**Table 3 nutrients-15-02986-t003:** Effect of nutrition education-only and yogurt-plus intervention on dietary diversity scores.

Dietary Diversity Indicators	Baseline	3-Month Follow Up	ITT Analysis N = 162		Baseline	3-Month Follow Up	Complete-Case Analysis N = 127	
	%	%	^§^ OR	95% CI	%	%	^§^ OR	95% CI
^‖^ Minimum dietary diversity (MDD)								
Control	14.81	54.90	Ref.	-	16.28	44.19	Ref.	-
Education-only	14.81	58.02	2.44	0.45, 13.21	19.05	64.29	2.62	0.43, 15.93
Yogurt-plus	07.41	59.48	5.00	0.79, 31.80	09.52	57.14	4.82	0.67, 34.86
^‖^ Minimum meal frequency (MMF)								
Control	33.33	89.54	Ref.	-	37.21	88.37	Ref.	-
Education-only	31.48	96.30	6.12	0.44, 85.70	30.95	97.62	9.65	0.65, 144.17
Yogurt-plus	38.89	98.69	-	-	45.24	100.00	-	-
^‖^ Minimum adequate diet (MAD)								
Control	11.11	45.75	Ref.	-	11.63	39.53	Ref.	-
Education-only	12.96	51.85	2.20	0.43, 11.20	16.67	64.29	2.24	0.40, 12.36
Yogurt-plus	05.56	62.09	5.72	0.89, 37.00	07.14	57.14	4.52	0.66, 31.16

^§^ Odds Ratio estimated using Logistic Mixed Model, missing data generated using multiple imputations. Missing estimates are due to convergence errors; ^‖^ MDD = consumption of ≥5 food group in past 24-h; MMF = consumption of solid or semi-solid foods >2 times in past 24-h; MAD = meeting both MDD and MMF.

## Data Availability

The data presented in this study are available on request from the corresponding author. The data are not publicly available due to privacy issues.

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
