# Peer review of "The Effects of Yogurt Supplementation and Nutritional Education on Malnourished Infants: A Pilot RCT in Dhaka’s Slums"

_nutrients, 2023, doi:10.3390/nu15132986_

Round 1

Reviewer 1 Report

1. The title should be changed.

2. The word "undernutrition" should be changed with  malnutrition. According to the WHO the proper term is malnutrition.

Please check the WHO website:

https://www.who.int/health-topics/malnutrition#tab=tab_1

3. Scheme 2. A calendar page filled out by a participating mother.- this figure should be removed.

4. Lines 187-188 " In addition to the above nutritional education activities, this group received a cup of 187 yogurt (50 gm)"

May be you mean 50 grams. Тhe abbreviation is not correct.The unit should be "g" or grams. 

https://fdc.nal.usda.gov/fdc-app.html#/food-details/2262074/nutrients

5. Materials and methods section should be included. The methods are not clearly described.

6. Please check the punctuation.

7. The text need English editing. You can check MDPI Editing services.

8. A conclusion must be included

9. References should be done according to MDPI guidelines.

Moderate editing of English language required.

Reviewer 2 Report

This is a pilot RCT study aimed to quantify the effects of yogurt supplementation and nutrition education over three 12 months on linear growth of infants at risk of stunting. The sample size was small. The protocol was sound. However, there were still some concerns for this study before publication.

1-      Does any review or meta-analysis related to this topic publish previously? The authors should add to introduction section.

2-      Per protocol and intention to treat should be considered for this pilot RCT.

3-      How is the difference between interim analysis and final analysis for study results?

4-      The odds ratio was unreal may be caused by sparse effect. (PMID: 34043251)

5-      Please give a rationale on figure 3.

6-      The references should be updated.

Round 2

Reviewer 1 Report

Dear authors,

the manuscript was definitely improved.

However, I have some remarks:

1. Scheme 1. Study sites at Dhaka, Bangladesh.-

The figure is unclear. Something more, do you have copyrights for this figure. In my view the figure must be replaced.

2. Scheme 2- must be removed. It reduces the scientific appearance of the manuscript.

3. The conclusion must be improved.

4. The references are not according to the MDPI guidelines.
